# Coating of Felt Fibers with Carbon Nanotubes and PEDOT with Different Counterions: Temperature and Electrical Field Effects

**DOI:** 10.3390/polym15204075

**Published:** 2023-10-13

**Authors:** Marta Carsí, María J. Sanchis, José F. Serrano-Claumarchirant, Mario Culebras, Clara M. Gómez

**Affiliations:** 1Department of Applied Thermodynamics, Instituto de Automática e Informática Industrial, Universitat Politècnica de Valencia, 46022 Valencia, Spain; 2Department of Applied Thermodynamics, Institute of Electrical Technology (ITE), Universitat Politècnica de València, 46022 Valencia, Spain; 3School of Engineering & Materials Science, Queen Mary University of London, Mile End Road, London E1 4NS, UK; 4Institute of Materials Science (ICMUV), Universitat de València, c/Catedràtic José Beltrán 2, 46980 Paterna, Spain; mario.culebras@uv.es

**Keywords:** PEDOT, MWCNT, felt, oxidative polymerization, electrical field

## Abstract

The use of wearable devices has promoted new ways of integrating these devices, one of which is through the development of smart textiles. Smart textiles must possess the mechanical and electrical properties necessary for their functionality. This study explores the impact of polymer-felt microstructure variations on their morphology, electrical, and mechanical properties. The application of thermal treatment, along with an electric field, leads to a substantial structural reorganization of the molecular chains within pristine felt. This results in a system of nanofibrils coated with MWCNT-PEDOT, characterized by highly ordered counterions that facilitate the flow of charge carriers. Both temperature and an electric field induce reversible microstructural changes in pristine felt and irreversible changes in coated felt samples. Furthermore, electropolymerization of PEDOT significantly enhances electrical conductivity, with PEDOT:BTFMSI-coated fabric exhibiting the highest conductivity.

## 1. Introduction

The recent development of portable, flexible, and thin electronics has promoted new ways of integrating these devices [1,2]. Consequently, a novel wearable technology has emerged, which aims to develop a new branch of products based on smart devices [3]. Specifically, devices based on smart fabrics, health monitoring systems, and wearable displays are becoming very popular due to their potential to transform several key industrial sectors such as: sports, health, and personal computing [4,5,6]. These applications require the development of new materials and designs capable of being integrated into fabrics adopting the flexibility, lightness, and softness necessary for being wearable [7,8]. Therefore, one the biggest challenges of wearable technology is the development of conductive fabrics that can easily integrate electronic circuits and be simultaneously part of the actual clothes. Numerous studies have been carried out on the development of electrically conductive textiles [9] for the manufacture of sensors [10], heaters [11], and power supplies [12]. The typical methodologies to produce conductive textiles are based on dip coating, the layer-by-layer technique and in situ oxidation by chemical or electrochemical polymerization.

However, only a few works have focused on considerations about the thermal management and mechanical properties necessary for the ergonomics of wearable devices [13]. Electronic components must be kept within certain temperature limits to ensure good performance. Furthermore, wearable devices are close to human skin and the surface of the device is required to be adequate [14]. Conventional polymers generally have low thermal conductivity, causing heat dissipation to be a bottleneck for device performance [13,15,16]. On the other hand, the integration of devices in wearable technology must ensure a certain level of comfort. Therefore, it is essential to study the mechanical flexibility of materials developed to be used in wearable electronics.

In previous works, we have successfully integrated different conductive polymers into woven and non-woven fabrics, previously coated with carbon nanotubes by means of electrodeposition [17,18]. We have also analyzed how the electrical conductivity of the prepared fabrics varies when applying different wear tests (torsion, bending), and we have made wearable thermoelectric generators (wTEG) with high output power [17,19]. Therefore, the next logical step towards the development of wTEG or, in general, wearable electronics based on fabrics coated with conductive polymers, is to study the influence of the coating on its mechanical properties.

Looking at this scenario, the present work shows an analysis of the thermal and mechanical behavior of uncoated and coated woven samples with MWCNT:PEDOT with different counterions and a study of how their dielectric properties change with changes in frequency and temperature under an electric field. This analysis constitutes a significant factor to be considered for woven fabrics in their electronic applications.

## 2. Materials and Methods

### 2.1. Materials

Poly(diallyldimethylammonium chloride) (PDADMAC) with a molecular weight of 10^5^ g mol^−1^, sodium deoxycholate (DOC), 1-Butyl-3-methylimidazolium hexafluorophosphate (PF_6_), and 1-Ethyl-3-methylimidazolium bis(trifluoromethylsulfonyl)imide (BTFMSI) were purchased from Sigma-Aldrich (Madrid, Spain). 3,4-Ethylenedioxythiophene (EDOT), lithium perchlorate, and acetonitrile were purchased from Alfa Aesar (Barcelona, Spain). Multiple-wall carbon nanotubes (MWCNTs) were obtained from Bayer Material Science (Leverkusen, Germany, 12–15 nm outer and 4 nm inner wall diameter, length > 1 mm, purity 95 wt%). Nonwoven felt fabric (made of polyester fibers) with a grammage of 140 g m^−2^ and 0.8 mm thickness was purchased from MW Materials World (Barcelona, Spain). All chemicals were used as received.

### 2.2. Sample Preparation

The felt fabrics coated with MWCNT and PEDOT with different counterions were prepared following the processes described in Figure 1 [17,19,20]. Coating over the felt fibers with MWCNT was carried out using the layer-by-layer technique (LbL), using PDADMAC and DOC as cationic and anionic stabilizer agents, respectively. This fabric was used as a working electrode for the electrochemical deposition of PEDOT:counterion. Three different solutions were used for the electrodeposition: 0.01 M EDOT and 0.1 M LiClO_4_; 0.01 M EDOT and 0.01 M PF_6_ and 0.01 M EDOT and 0.01 M BTFMSI for the synthesis of PEDOT:ClO_4_, PEDOT:PF_6_; and PEDOT:BTFMSI, respectively. Three electrodes formed the electrochemical cell: a counter electrode (platinum grid), a reference electrode (Ag/AgCl), and the MWCNT–fabric as the working electrode where a constant current intensity of 6 mA was applied for 1 h.

### 2.3. Morphological Characterization (SEM)

Morphological characterization was carried out using a Hitachi 4800 S (Tokyo, Japan) field-emission scanning electron microscope (SEM) at an accelerating voltage of 20 kV and a working distance of 14 mm for palladium-gold-coated surfaces.

### 2.4. Thermogravimetric Analysis (TGA)

TGA measurements were performed on a TGA 550 from TA Instruments (TA Instruments, Cerdanyola del Valles, Spain) using platinum pans under a 50 mL min^−1^ flow of air. The measurement was performed from 20 to 700 °C, using a rate of 10 °C min^−1^.

### 2.5. Differential Scanning Calorimetry (DSC)

The thermal program used consisted of a first heating cycle up to 300 °C to erase the thermal history of the sample. Then, the samples were kept at 300 °C for 10 min to ensure that all the samples were molten. After this first heating cycle, a cooling process was carried out to 0 °C at 50 °C min^−1^, and finally, a second heating cycle was carried out to 300 °C at 10 °C min^−1^.

### 2.6. Dynamo-Mechanical Analysis (DMA)

The dynamic mechanical properties of samples of rectangular shape were evaluated by dynamical mechanical analysis (Q800 DMA, TA Instruments) in tension mode (preload force; 0.05 N) with a temperature ramp of 1 °C min^−1^. Measurements for all the samples were taken under the same conditions from room temperature to 180 °C at a frequency of 1 Hz.

### 2.7. Analysis by Dielectric Relaxation Spectroscopy (DRS)

DRS measurements in a frequency range from 5·10^−2^ to 3·10^6^ Hz were performed using a Novocontrol Broadband Dielectric Spectrometer (Hundsagen, Germany) consisting of an SR 830 lock-in amplifier with an Alpha dielectric interface. The measurements were performed in an N_2_ atmosphere from 20 °C to 160 °C and to 100 °C, in steps of 5 °C, for the pristine and coated felt samples, respectively. Temperature control was carried out by means of a Novocontrol Quatro cryosystem, with an accuracy of ±0.1 °C during each sweep in frequency. Disc-shaped samples, of about 0.1 mm thickness and a 20 mm diameter, were used. The experimental uncertainty was better than 5% in all cases.

## 3. Results

The homogeneity of the coating of felt fabrics with MWCNTs and PEDOT with different counterions was determined by scanning electron microscopy (SEM). Figure 2 shows SEM micrographs of the samples with and without different coatings.

On the one hand, after coating the felt fabric with multi-walled carbon nanotubes, fibers are perfectly coated with MWCNTs and some PDADMAC-DOC agglomerates (Figure 2b). Once the felt fabrics have been coated with MWCNTs, PEDOT is electrodeposited in the presence of different counterions. At first glance, a clear trend is observed, indicating that PEDOT electrodeposition using ionic liquids as counterions leads to a higher degree of fiber coating compared to fibers coated with lithium perchlorate as a counterion. Moreover, several differences in terms of the morphology of the coating were observed between the different samples produced. For the PEDOT:ClO_4_ (Figure 2c), the coating depicted a globular morphology, whereas the coating based on PEDOT:PF_6_ (Figure 2d) and PEDOT:BTFMSI (Figure 2e) presented a cauliflower-like morphology indicating a more compacted deposition which is preferable for electric transport. Both morphologies are typical of the electrochemical deposition of conductive polymers by chronoamperometry [21,22].

The properties of textiles can be affected when they are heated due to their thermosensitivity. Therefore, analysis of thermo-mechanical properties is required to evaluate the use of the fabrics in wearable devices after the deposition of the different coatings on the felt fibers. The thermogravimetry curves (TGA) in Figure 3 show that the degradation of the felt control sample takes place in two steps, the first with a thermogravimetric-derived temperature peak of 425 °C and the second with a peak of 480 °C [23]. When the felt fibers are covered with carbon nanotubes, a slight increase in thermal stability is observed because the decomposition initiation temperature (T_5%_) increases by 10 °C, see Table 1. However, when the fibers are electrochemically coated with PEDOT, a decrease in mass is observed at temperatures below 367 °C. This weight loss in the range between 300 and 400 °C could be attributed to the decomposition of the thiophene chain [24].

In addition, the decomposition initiation temperature decreases by approximately 100 °C for PEDOT:ClO_4_, 50 °C for PEDOT:PF_6_, and 40 °C for PEDOT:BTFMSI compared to the initiation temperature of felt fabric decomposition, as seen in Table 1. When comparing different PEDOT:counterion coatings, it can be observed that PEDOT coatings with ionic liquids as counter ions provide greater thermal stability than the PEDOT:ClO_4_ coating. Therefore, these fabrics are more stable against higher thermal gradients.

Differential scanning calorimetry (DSC) allows the rapid detection and measurement of physical and chemical transformations in the material as a function of temperature. The thermogram obtained from the second heating cycle, Figure 4, was used for the analysis. It shows a broad step in the baseline of the measurement curve around 80 °C associated with the glass transition of the felt. At higher temperatures, an endothermic process is observed between 240–255 °C associated with the melting of the crystalline phase present in the samples. The thermogram corresponding to the second heating cycle of the pristine felt fabric shows a glass transition temperature (*T_g_*) at 78 °C and a melting point of 253 °C. The DSC thermogram of pure polyethylene terephthalate (PET) shows a *T_g_* at 85.4 °C and a *T_m_* of 254.9 °C with good agreement with the felt fabric employed [25,26].

Focusing on the evolution of the *T_g_* after the different coatings, we can observe that it decreases when the felt fibers are coated with the MWCNTs, and it decreases even more after the electrodeposition of PEDOT with the different counterions. Especially noticeable is the case of the use of ionic liquids as counterions of the PEDOT chains because the glass transition temperature decreases by 10 °C compared to the pure felt fabric. This suggests that the ionic liquid can easily diffuse on the surface of the fibers resulting in an improved plasticization of the felt fibers [27]. It can also clearly be seen in the DSC thermogram that adding the different coatings decreases the melting temperature while keeping the melting enthalpy constant (Table 2). This melting point decrease indicates that the coating of felt fibers slightly reduces backbone rigidity. From the values of the enthalpy of fusion, Δ*H_m_* (J g^−1^), obtained by measuring the area under the melting curve in each of the thermograms, we can estimate the crystalline fraction *χ_c_* of each sample using Equation (1). We compared the values obtained with the crystalline melting enthalpy for a 100% crystalline polyester felt material. For the calculation, a value of Δ*H_m,lit_* = 140.1 J g^−1^ was used in accordance with the literature [26].
(1)$$\chi_{c}\left(\%\right)=\frac{\Delta H_{m}}{\Delta H_{m,\;lit}}\cdot 100$$

According to Rodríguez et al. [28], polyester fibers are obtained by spinning the polymer above its glass transition temperature. This process produces an outer layer in the fibers whose orientation is significantly greater than that of the inner or core layer. So, the felt fibers are composed of microfibrils with crystalline blocks (crystallites) linked by an isotropic amorphous phase oriented in the fiber axis direction. These microfibrils are linked to each other in the transverse direction by an amorphous phase also oriented along the fiber axis. Figure 5 shows the pattern of the felt fiber microstructure.

The dynamic mechanical analysis provides valuable information about the polymer stiffness, molecular motion, and relaxation processes of composite systems [29]. Figure 6 shows the evolution of the storage modulus (*E*′), loss modulus (*E*″), and tan δ as a function of temperature for the felt, felt + MWCNT, and felt + MWCNT coated with PEDOT:ClO_4_ fabrics, PEDOT:PF_6_ and PEDOT:BTFMSI by electrodeposition. From the DSC and DMA results, it is clear how the complex microstructure of the polyester fibers affects molecular mobility, both in terms of the amorphous phase portions and the crystalline phase portions present. The plot of tan δ and loss modulus (*E*″) as a function of temperature shows the presence of two relaxation processes attributed to two glass transition temperatures. This indicates the presence of two amorphous phases of a different nature, with very different mobility in the chain segments. The first, around 80 °C, corresponds to the free amorphous phase with greater mobility. The second, around 125 °C, corresponds to the amorphous phase constrained between the crystalline phase of the microfibrils that make up the felt polyester fibers, with restricted mobility. This second glass transition is very poorly visible in DSC thermograms.

In addition, Figure 6 shows that the pristine felt fabric has a storage modulus value at 35 °C of 3.2 MPa. After the deposition of 20 bilayers (BL) of MWCNT by LbL on the felt fabric, the storage modulus at 35 °C increased to 38 MPa, indicating that MWCNT acts as a reinforcement [30]. However, the electrochemical polymerization of PEDOT with the different counterions reduced the storage modulus at 35 °C until 15 MPa, 4.7 MPa, and 3.5 MPa for the PEDOT:PF_6_, PEDOT:BTFMSI, and PEDOT:ClO_4_ coatings, respectively.

The nature of the felt fabric is non-woven, and therefore the felt fibers are randomly distributed and spaced. The behavior of the storage modulus can be attributed to the fact that during the coating of the felt fibers with MWCNT, the fibers compact, decreasing the free volume between them. However, upon electrodeposition of PEDOT with the different counterions, the PEDOT polymer chains begin to grow on the surface of the MWCNT-coated fibers so that the free volume between the fibers increases. These results show that coating the felt fibers with carbon nanotubes increases viscoelastic stiffness. However, the electrodeposition of PEDOT with different counterions decreases, and the PEDOT:ClO_4_ and PEDOT:BTFMSI coatings have similar viscoelastic stiffness to that of pristine felt. Moreover, the tan δ shows two relaxation processes corresponding to the different types of molecular arrangement present in the microfibrils that make up the felt. As mentioned above, the felt has two types of amorphous phases. One amorphous phase is confined between crystalline blocks and acts as a connecting element in the fiber axis direction between the crystalline blocks. The other amorphous phase, preferentially oriented along the fiber axis, is a transversal junction with the other amorphous phase confined between the crystalline blocks. The decrease in tan δ values at 35 °C after adding different coatings to the felt fibers suggests a good interfacial adhesion between the fibers and the coatings [27,31].

Dielectric spectroscopy is a non-destructive method providing information about the reorientation and rotations of the main and segmental chains’ dipoles and conductivity mechanisms. This technique allows the characterization of both dipole and conductive processes as a function of frequency and temperature. Electrical properties are related to polymer-felt microstructure variations. Consequently, a study of the dielectric properties of felt has been developed. The real component of the complex permittivity represents the ability of a material to store electric charge under the influence of an electric field, and its value is strongly dependent on the structure. Figure 7 shows the evolution of the dielectric constant (ε′) and the loss factor (ε″) as a function of temperature at 1 Hz for the felt sample. The relative permittivity of felt is very close to one (air) due to its porosity (~40% air in volume), and its value increases with increasing temperature up to its maximum value around 50 °C and then reduces up to approximately 90 °C, from which temperature its value increases again. Again, the results show that a change in the arrangement of molecular chains, that is, in the felt microstructure, occurs at this temperature range.

This effect is also evident in mechanical measurements, where an increase in modulus is observed up to 60 °C, followed at temperatures above this value by a reduction as expected for an increase in mobility with increasing temperature (see inset Figure 7). The rise observed by increasing temperature is widespread in polycrystalline materials [32]. An increase in the ordering of molecular chains present translates into an increase in the mechanical modulus and a reduction in the dielectric constant values. This is related to increased hindrance to the flow of charge carriers. On the other hand, the temperature dependence of the loss permittivity shows the presence of two closely overlapping dipole processes in the temperature range analyzed, according to the DMA results. The low- and high-temperature processes are associated with the segmental relaxation of the non-confined and confined amorphous phase, respectively.

Recent studies have shown that the mechanical and dielectric properties of coated fiber can be modified due to changes in the molecular morphology that occur in its processing and/or use conditions [33,34,35]. Considering this effect, we have evaluated the effect on the dielectric response after applying an electrical field in consecutive sweeps at a constant temperature (25, 50, and 75 °C). Figure 8 shows the dielectric loss modulus frequency dependence of five consecutive scans at 75 °C in the experimental frequency range. As we can see, a reduction in height and width is observed when repeating the scan. The process associated with the glass transition temperature becomes narrower and lower in intensity. These changes were not observed at 25 °C and 50 °C. These results show that at 75 °C, there is a change in the felt microstructure. Specifically, an increase in the most orderly phase present translates into immobilizing some molecular chains participating in the dipolar process associated with the glass transition temperature. DRS experiments have shown that this is the characteristic evolution of the α-relaxation during a molecular chain ordering process [36].

The behavior of the felt support with temperature changes will determine the behavior of the coated felt samples with carbon nanotubes and with PEDOT with different counterions. As we are interested in the conductivity of the samples for their application as materials in thermoelectric applications, we measured the dependence of the conductivity of the felt support as a function of temperature, as well as the dependence of the conductivity with the composition of the coated felt samples. As shown in Figure 9, the conductivity of the felt sample increases with temperature until it reaches a temperature of 125 °C, reducing its value afterward. This temperature coincides with the glass transition temperature for the amorphous fraction confined between the ordered phases [28,37]. The results seem to indicate that the movement of the previously restricted molecular chains exerted a negative effect on the conductive process. This result implies that a significant structural transition occurs in the felt samples as the temperature reaches the glass transition value for the amorphous phase. Considering this effect and the temperature range in which these samples would potentially be used, the temperature dependence of the conductivity for the coated samples between 25 °C and 100 °C was analyzed (Figure 10). As expected, when carbon nanotubes cover the felt, the dipolar processes of the samples are not visible because the conductive processes mask them [38]. For all the samples in the analyzed temperature range, an increase in conductivity was observed with increasing temperature, except for the felt+MWCNT-PEDOT: BTFMSI sample, in which a slight reduction in conductivity was observed after reaching 90 °C.

Taking this effect into account and the range of temperatures in which these samples would potentially be used, the frequency dependence of the conductivity of the samples was analyzed at three different temperatures, 25, 50 and 75 °C. In Table 3, the dc conductivity values for the five samples at these three temperatures are summarized. These values were obtained from the plateau at low frequencies.

As the felt substrate changes its dipolar response after performing consecutive cycles (see Figure 8), it is necessary to re-examine whether the conductive properties of pristine felt and coated felt with nanotubes and PEDOT with different counterions are desirable after being subjected to different cyclical conditions. For this purpose, consecutive thermal cycles conductivity measurements were performed for each one of the samples. Figure 11 shows the results obtained for the felt, felt coated with MWCNT, and after PEDOT deposition with different counterions: PF_6_, and BTFMSI. Results demonstrate that although the conductivity values of the pristine felt sample were not significantly affected by the consecutive measurement cycles performed, there was remarkable variation in the conductivity of the coated felt samples after the electric field was applied. It was observed that the conductivity changes were greater when the temperature was higher than 60 °C. These results indicate that the structural reorganization that occurs in the felt substrate above 60 °C favors the redistribution of the coating, which translates into an irreversible increase in the conductivity values.

Studies have shown that the alignment of MWCNTs occurs in several stages. In the first, they are oriented in the direction of the field. In the second stage, the polarized MWCNTs attract each other forming aggregations in a chain-like structure. This reorganization provides a direct path of flow, which can lead to fantastic enhancement in electrical conductivity [35].

Normally, when the electric field is removed, the Brownian diffusion is relevant to the motion, and the aligned network tends to fade until it returns to the primary random distribution situation. However, the felt-coated samples analyzed show that conductivity increases at a constant temperature cycle after cycle, and does not recover the initial values after reducing the temperature, probably due to the irreversible orientation of the MWCNT forced by the microstructure change in the felt substrate on which they are deposited.

The applied voltage led to the alignment of MWCNTs inside the felt substrate, whereas the current generated as the material’s response to the applied field favored the alignment of MWCNTs by joining them to be end-to-end. Although conductivity increases with temperature, this rise decreases after several temperature cycles. That is, after having subjected the sample to an electric field at 75 °C, the conductivity measured at 25 °C presents values very close to those obtained at 75 °C. It is as if the microstructural changes produced favor a direct flow path, which represents a fantastic improvement in electrical conductivity over a greater range of temperatures. The results provide a general pathway for the controlled assembly of nanomaterials and organized devices.

## 4. Conclusions

Preparing valuable materials and functional devices requires devising processing strategies that enable the controlled assembly of material components into structures with a designed order. Post-synthesis strategies for organizing constituents include self-assembly driven by specific interactions between constituents using different procedures. In our study, it has been confirmed how the different nature of the constituents, as well as the experimental conditions chosen, influence the properties and therefore performance of the materials obtained.

According to our results, the glass transition temperature, *T_g_*, of coated felt samples is lower than that of uncoated felt, which contributes to enlarging the service temperature range of felt. The coating of felt fibers with carbon nanotubes increases the viscoelastic stiffness. However, after the electrodeposition of PEDOT with different counterions it decreases, having a similar viscoelastic stiffness to that of pristine felt. Moreover, after the electropolymerization of PEDOT, the electrical conductivity increased more than three orders of magnitude, the most conductive being the fabric with PEDOT:BTFMSI.

The microstructure of both pristine and coated felt samples changes with the temperature and under the effect of the electric field. Both the applied voltage and reorganization chain (microstructure changes) of the felt substrate can highly affect the alignment process, allowing the maintenance of high conductivity values in a wide range of temperatures. These results make the materials more attractive for their application in thermoelectric processes.

Likewise, the temperature induces reversible microstructural changes in electrical conductivity in the pristine felt, but irreversible changes in the coated felt samples when an electric field is applied cyclically. This interesting result indicates that the microstructure of coated felt samples changes under the effect of the electric field. The electric field favors the interactions of the deposited materials with the felt substrate.

The materials obtained are easily processable, lightweight, flexible and highly conductive, in particular the fabric coated with PEDOT:BTFMSI. These advantages open the gateway for the application of these hybrid materials in the modem electronic industry in which device integration into portable goods is gaining more importance

## Figures and Tables

**Figure 1 polymers-15-04075-f001:**
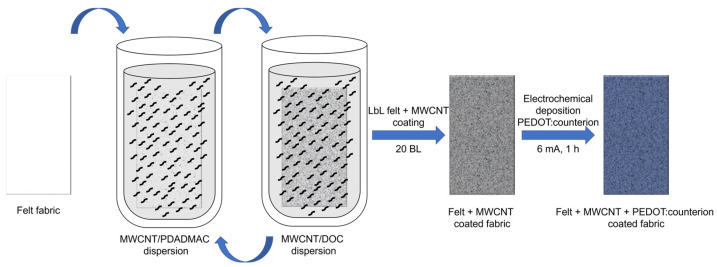
Scheme of the LbL felt + MWCNT coating process and electrochemical deposition of PEDOT: counterions.

**Figure 2 polymers-15-04075-f002:**
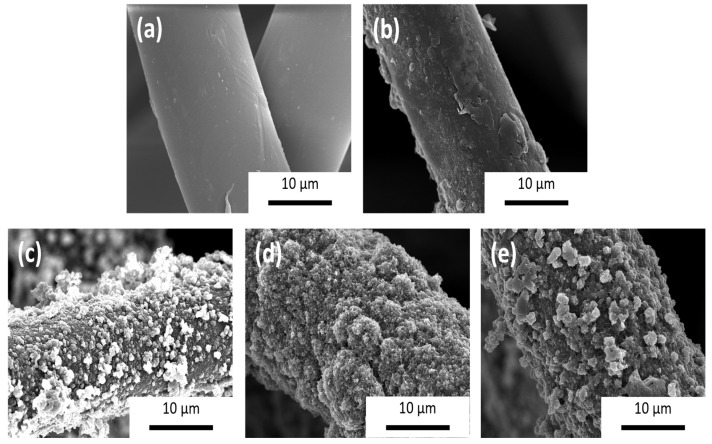
SEM images of felt (**a**), felt-coated fibers with multi-walled carbon nanotubes (MWCNT) (**b**), and felt coated with MWCNT and PEDOT:ClO_4_ (**c**); PEDOT:PF_6_ (**d**) and PEDOT:BTFMSI (**e**).

**Figure 3 polymers-15-04075-f003:**
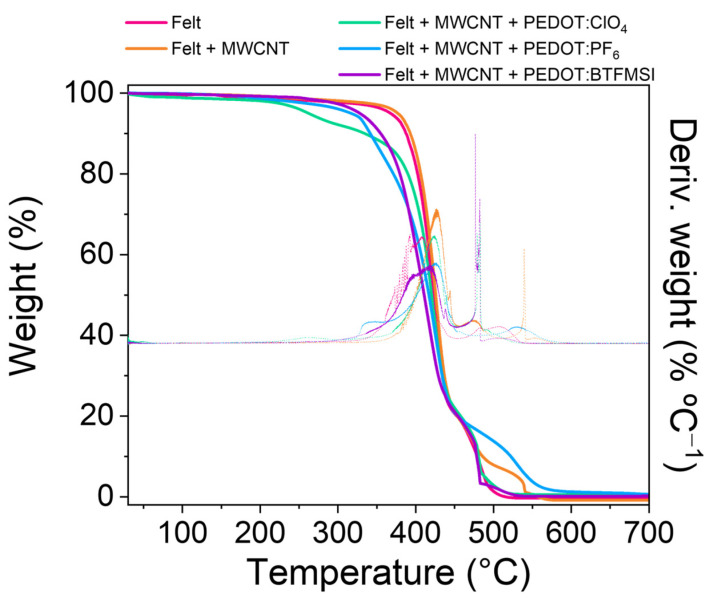
TGA measurements of felt fabric and coated felt fabrics.

**Figure 4 polymers-15-04075-f004:**
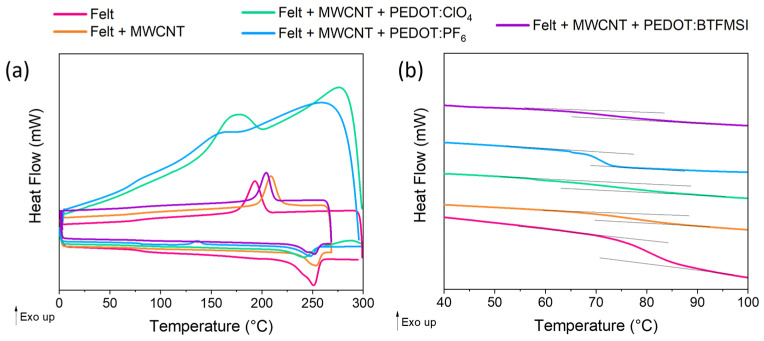
(**a**) DSC measurements of felt, felt coated with MWCNT, and after PEDOT deposition with different counterions: PEDOT:ClO_4_, PEDOT:PF_6_, and PEDOT:BTFMSI. (**b**) Magnified thermogram between 40 and 100 °C, where the *T_g_* appears. The thermograms in (**b**) have been moved vertically to better observe the evolution of the glass transition temperature.

**Figure 5 polymers-15-04075-f005:**
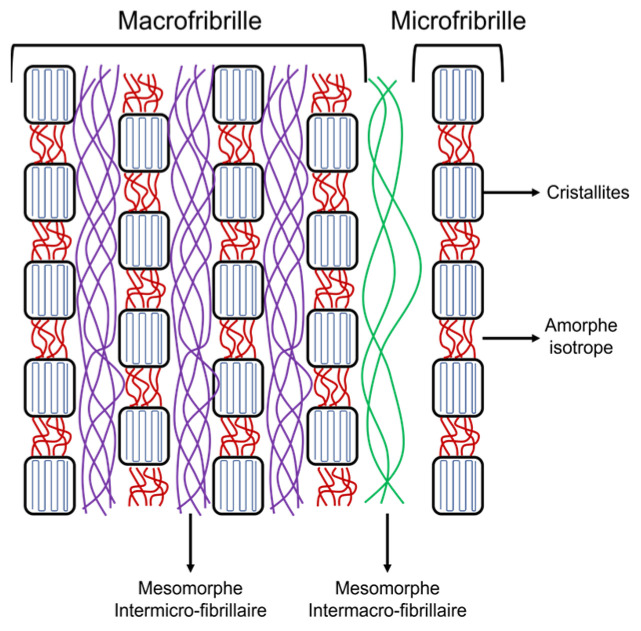
Model of felt fiber microstructure.

**Figure 6 polymers-15-04075-f006:**
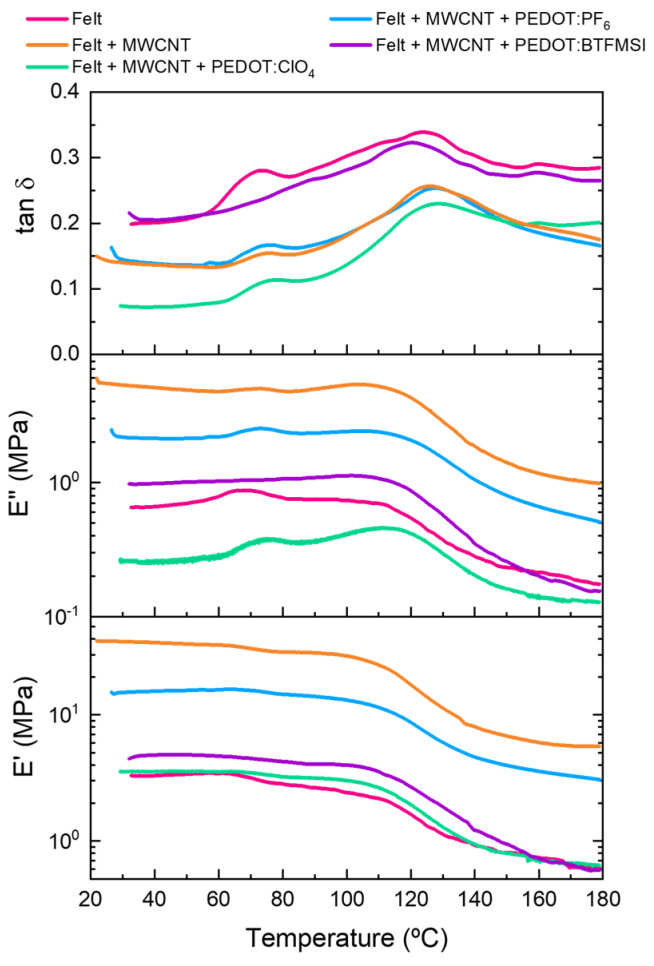
Storage modulus (*E*′), loss modulus (*E*″), and tan δ as a function of the temperature for the felt, felt coated with MWCNT, and after PEDOT deposition with different counterions: PEDOT:ClO_4_, PEDOT:PF_6_, and PEDOT:BTFMSI.

**Figure 7 polymers-15-04075-f007:**
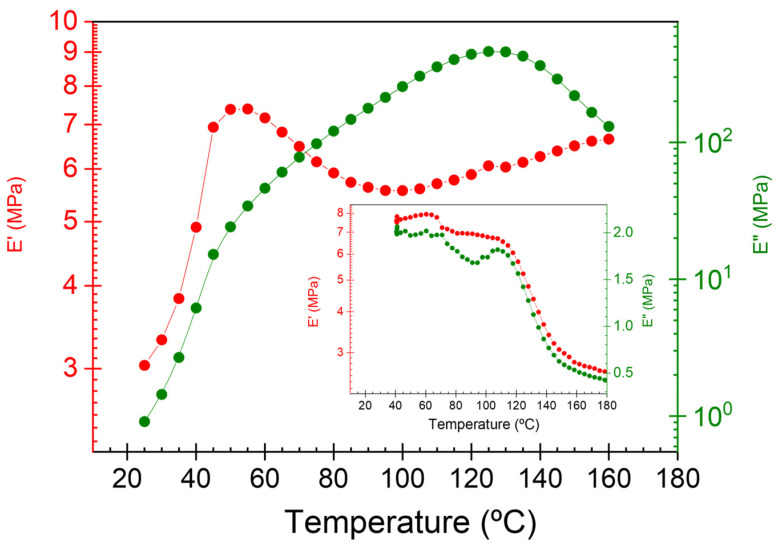
Temperature dependence of the real and imaginary component of the complex permittivity at 1 Hz for the felt sample. Inset shows the temperature dependence of the real and imaginary component of the mechanical modulus at 1 Hz.

**Figure 8 polymers-15-04075-f008:**
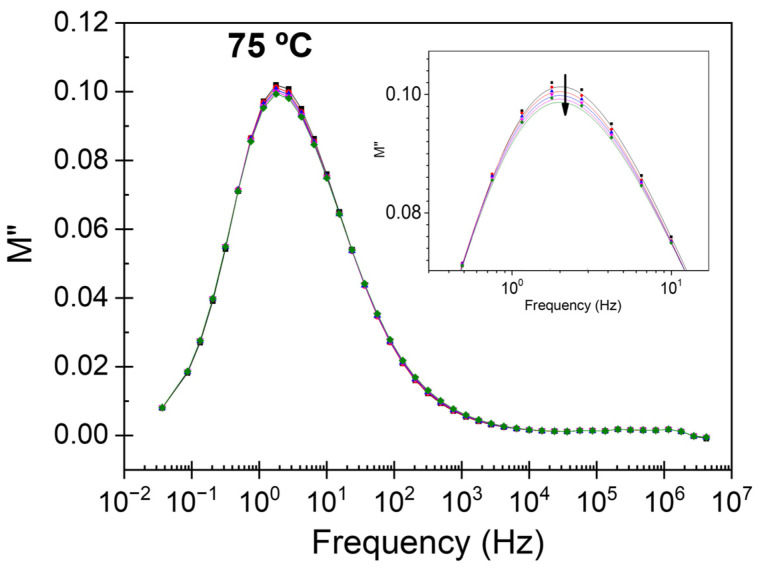
Frequency dependence of loss dielectric modulus at 75 °C for the felt sample. Inset shows the zoom of the frequency area of the process.

**Figure 9 polymers-15-04075-f009:**
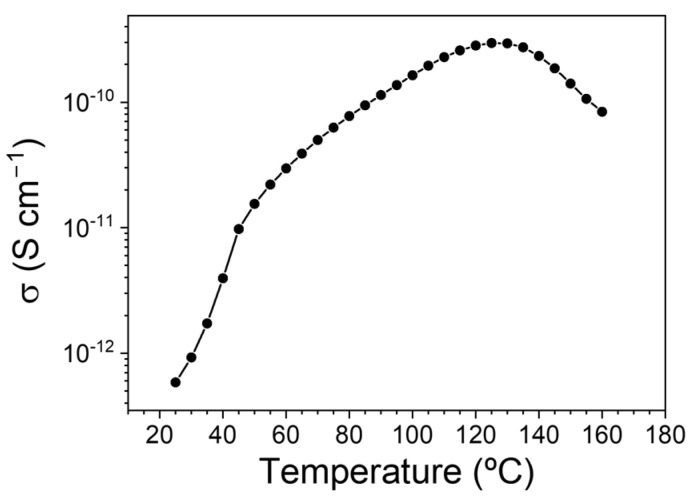
Temperature dependence of the conductivity at 1 Hz for the felt sample.

**Figure 10 polymers-15-04075-f010:**
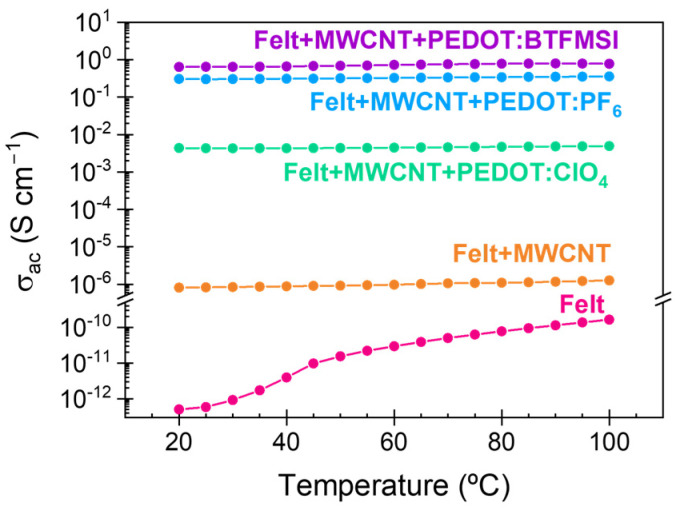
Temperature dependence of the conductivity of felt, felt coated with MWCNT, and after PEDOT deposition with different counterions: ClO_4_, PF_6_, and BTFMSI.

**Figure 11 polymers-15-04075-f011:**
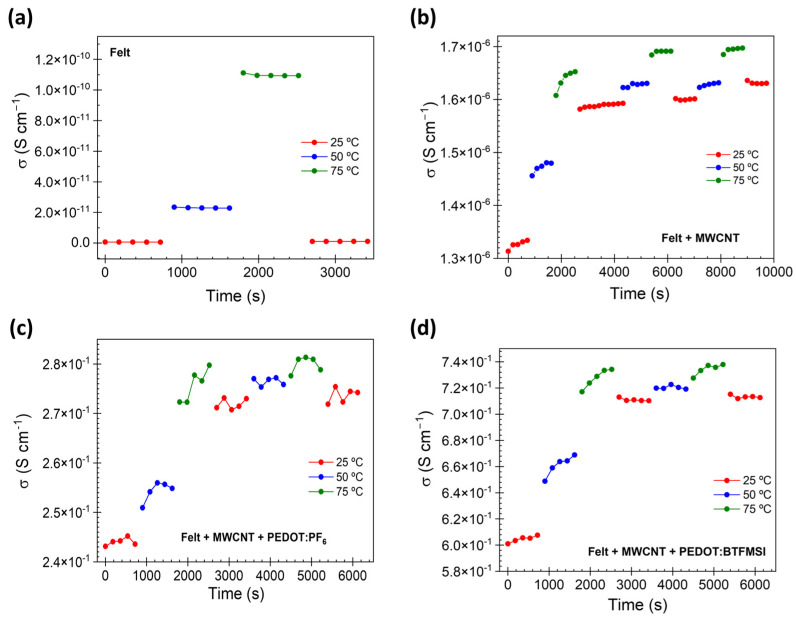
Electrical conductivity after electric voltage is applied cyclically (1V) for: (**a**) felt, (**b**) felt coated with MWCNT, (**c**) felt coated with MWCNT and PEDOT:PF_6_, and (**d**) felt coated with MWCNT and PEDOT:BTFMSI.

**Table 1 polymers-15-04075-t001:** Weight loss temperature at different percentages and residue percentages at 700 °C of pristine felt fabric and felt fabric coated with MWCNT, MWCNT + PEDOT:ClO_4_, MWCNT + PEDOT:PF_6_*,* and MWCNT + PEDOT:BTFMSI.

	Felt	Felt + MWCNT	Felt + MWCNT + PEDOT:ClO_4_	Felt + MWCNT + PEDOT:PF_6_	Felt + MWCNT + PEDOT:BTFMSI
T_5%_ (°C)	367.65	376.85	262.22	316.87	328.10
T_10%_ (°C)	387.51	393.02	334.41	341.50	354.88
T_20%_ (°C)	402.33	406.45	389.81	371.16	379.62
T_30%_ (°C)	411.02	414.41	404.90	393.11	391.97
T_40%_ (°C)	417.45	420.40	413.88	406.97	401.68
T_50%_ (°C)	422.87	425.65	420.59	416.94	410.81
T_60%_ (°C)	427.91	430.68	426.71	425.74	417.80
T_70%_ (°C)	433.93	436.84	434.19	433.91	429.01
T_80%_ (°C)	452.84	453.28	457.64	454.26	454.06
T_90%_ (°C)	479.08	486.04	480.68	522.32	478.32
T_95%_ (°C)	487.97	530.56	488.68	543.47	482.07
Residue % (700 °C)	0	0	0.56	0.615	0.199

**Table 2 polymers-15-04075-t002:** Glass transition temperature (*T_g_*), melting temperature (*T_m_*), enthalpy of fusion (Δ*H_m_*) and crystallinity percentage (*χ_c_*) of felt fabric and coated felt obtained by DSC.

Sample	*T_g_* (°C)	*T_m_* (°C)	$\Delta H_{m}$ (J g^−1^)	$\chi_{c}\;(\%)$
Felt	78.64	252.96	45.493	32.5
Felt + MWCNT	76.03	246.58	43.581	31.1
Felt + MWCNT + PEDOT:ClO_4_	73.34	242.90	51.004	36.4
Felt + MWCNT + PEDOT:PF_6_	68.09	242.09	46.695	33.3
Felt + MWCNT + PEDOT:BTFMSI	68.85	240.28	45.990	32.8

**Table 3 polymers-15-04075-t003:** Conductivity at 25, 50 and 75 °C of felt fabric and coated felt obtained by DRS.

	σ (S cm^−1^)		
Sample	25 °C	50 °C	75 °C
Felt	6.8714 × 10^−13^	2.3347 × 10^−11^	1.0412 × 10^−10^
Felt + MWCNT	8.2385 × 10^−7^	9.1966 × 10^−7^	1.0794 × 10^−6^
Felt + MWCNT + PEDOT:ClO_4_	4.3024 × 10^−3^	4.3714 × 10^−3^	4.6916 × 10^−3^
Felt + MWCNT + PEDOT:PF_6_	2.4313 × 10^−1^	3.1838 × 10^−1^	3.4121 × 10^−1^
Felt + MWCNT + PEDOT:BTFMSI	6.4476 × 10^−1^	6.8881 × 10^−1^	7.6804 × 10^−1^

## Data Availability

The data presented in this study are available on request from the corresponding author.

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
