# Peer review of "Coating of Felt Fibers with Carbon Nanotubes and PEDOT with Different Counterions: Temperature and Electrical Field Effects"

_polymers, 2023, doi:10.3390/polym15204075_

Round 1

Reviewer 1 Report

This paper discusses the influence of the thermal and mechanical behaviour of the uncoated and coated woven samples with MWCNT:PEDOT with different counterions, and also how their dielectric properties change with the changes in frequency and temperature under an electric field.  Overall, this is a clear, concise, and well-written manuscript. The introduction is relevant and grounded in theory. Sufficient information on the findings of the previous study is presented to enable readers to follow the rationale and procedures of this study. The results are clear and convincing. The publication is recommended for publication.

Author Response

This paper discusses the influence of the thermal and mechanical behaviour of the uncoated and coated woven samples with MWCNT:PEDOT with different counterions, and also how their dielectric properties change with the changes in frequency and temperature under an electric field. Overall, this is a clear, concise, and well-written manuscript. The introduction is relevant and grounded in theory. Sufficient information on the findings of the previous study is presented to enable readers to follow the rationale and procedures of this study. The results are clear and convincing. The publication is recommended for publication.

We appreciate the reviewer's feedback and their recommendation for publication.

Reviewer 2 Report

1. The Abstract is too brief, which should include important conclusions, or mechanisms given by this article based experimental data.

2. It will be better that the illustration of LbL coatings process was shown in this article.

3. Was the felt (fabric) applied here nonwoven or woven?

4. Why not the PDADMAC, DOC shown in the 2.1 Mateirals were used in any sample preparation or characterization procedures.

5. The loading amount of every coating should be given, which may be affected by the different counterions? Why did counterion of BTFMSI show the best conductivity?

6. How about the fastness of these coatings?

7. It needs more evidences, and not just a Model in Figure 4, such as, XRD, TEM, XPS,

Good.

Author Response

  1. The Abstract is too brief, which should include important conclusions, or mechanisms given by this article based experimental data.

Thank you for your suggestion. The abstract has been modified, including the most relevant findings of this study.

"The use of wearable devices has promoted new ways of integrating these devices, one of which is through the development of smart textiles. Smart textiles must possess the mechanical and electrical properties necessary for their functionality. This study explores the impact of polymer-felt microstructure variations on their morphology, electrical, and mechanical properties. The application of thermal treatment, along with an electric field, leads to a substantial structural reorganization of the molecular chains within the pristine felt. This results in a system of nanofibrils coated with MWCNT-PEDOT, characterized by highly ordered counterions that facilitate the flow of charge carriers. Both temperature and an electric field induce reversible microstructural changes in pristine felt and irreversible changes in coated felt samples. Furthermore, electropolymerization of PEDOT significantly enhances electrical conductivity, with PEDOT:BTFMSI-coated fabric exhibiting the highest conductivity."

  1. It will be better that the illustration of LbL coatings process was shown in this article.

Thank you for your suggestion. A schematic procedure of the LbL coating of MWCNT and electrochemical deposition of PEDOT:counterion has been included (Figure 1). The rest of the figure captions have been modified accordingly.

  1. Was the felt (fabric) applied here nonwoven or woven?

The felt fabric used in this article is nonwoven. This characteristic has been specified in section 2.1 Materials.

  1. Why not the PDADMAC, DOC shown in the 2.1 Materials were used in any sample preparation or characterization procedures.

PDADMAC and DOC are stabilizing agents for the colloidal dispersions of multi-walled carbon nanotubes (MWCNTs). These agents also impart a positive (PDADMAC, cationic polyelectrolyte) or negative (DOC, anionic surfactant) surface charge to the MWCNTs, which is necessary to carry out the LbL deposition through electrostatic interaction between these agents. In section 2.2 Sample preparation, both stabilizing agents are mentioned.

  1. The loading amount of every coating should be given, which may be affected by the different counterions? Why did counterion of BTFMSI show the best conductivity?

After the deposition of carbon nanotubes onto the felt fabric through LbL, the mass of the felt fabric increases by 70 mg, corresponding to a 45% increase. Following the electrochemical deposition of PEDOT with different counterions, the mass of the MWCNT-coated fabric increases on average by 25 mg, equivalent to an 11% increase. Preliminary studies showed that the deposition of PEDOT:ClO4 led to a fabric mass increase of around 22 mg. However, when using the counterions PF6 and BTFMSI, the fabric mass ranged between 28 and 32 mg.

The electrical conductivity of conducting polymers depends on various factors, the most important being the concentration of charge carriers and the morphology of polymer chains. In a previous work [ref. 19], we discussed the influence of counterions on electrical conductivity. Since the electrochemical deposition method is the same for different PEDOT:counterion coatings (chronopotentiometry, 6 mA, 60 min), the concentration of charge carriers is approximately the same for all PEDOT:counterion coatings. This was confirmed through Hall effect measurements (Figure 4b in ref. [19]). However, these measurements showed a substantial increase in carrier mobility. Since carrier mobility is inherently related to the morphology of the conducting polymer, it was determined that the higher electrical conductivity of the BTFMSI counterion is due to its larger size compared to other counterions, inducing a linear morphology in the PEDOT polymer chains, allowing for better alignment and, consequently, higher carrier mobility.

  1. How about the fastness of these coatings?

In this article, fastness tests for the different coatings on the fabrics have not been conducted as it was not the focus of this study. However, we consider including these tests in future work due to their significance in developing wearable textile electronics. On the other hand, in previous studies (refs. [17-19]), we conducted torsion and bending tests, which demonstrated that the PEDOT coatings remained stable and adhered to the textile fibres after wear tests. These tests were carried out by monitoring changes in electrical conductivity.

  1. It needs more evidences, and not just a Model in Figure 4, such as, XRD, TEM, XPS,

We believe that the thermal characterization performed with DMA, a particularly sensitive technique,  provides sufficient evidence of the two unambiguous glass transitions (Tg), indicating the presence of two amorphous phases of different nature with significantly different chain segment mobility.

Reviewer 3 Report

The article “Coating of felt fibers with carbon nanotubes and PEDOT with different counterions: temperature and electrical field effects” describes an approach to obtaining woven materials with specified electrical properties. The paper undoubtedly has practice interest.  But there are some questions, some of them listed below:

11.      Frequency selected for DMA is 1 Hz. Authors should justify their choices.

22.      DRS measurements were carried out in the temperature range from 20 to 160 degrees. The authors should justify the selected temperature range.

33.      The authors should clarify the adhesion of the coating to the fiber. This has important practical implications. When using coated fiber, some of the coating may come off due to poor adhesion.

44.      The authors should clarify paragraph lines 147-151. From Table 1 it follows that the decomposition temperature does not decrease for all coatings. A decrease in decomposition temperature leads to a decrease in thermal stability.

55.      The plasticity of felt fibers does not depend on the ionic liquid on their surface. The authors should clarify lines 188-190

66.      The statement (lines 220-221) is doubtful. Fiber coating is not taken into account.

77.      Lines 233-235: does this mean the destruction of the MWCNT coating after electrochemical polymerization with various ions.

88.      Graphs 8 and 9 for felt are different. The authors should correct these graphs.

Author Response

The article "Coating of felt fibers with carbon nanotubes and PEDOT with different counterions: temperature and electrical field effects" describes an approach to obtaining woven materials with specified electrical properties. The paper undoubtedly has practice interest.  But there are some questions, some of them listed below:

  1. Frequency selected for DMA is 1 Hz. Authors should justify their choices.

To study the mechanical response of the samples and determine their glass transition temperature (Tg), Dynamic Mechanical Analysis (DMA) tests were performed.

The standard experiment to determine Tg via DMA is to ramp the temperature of a sample while applying a small amplitude linear oscillation to measure the dynamic moduli E', E'', and tand. The glass transition temperature (Tg) depends on the polymer but is influenced by other factors. In this sense, the measured Tg is quite sensitive to the oscillation frequency, obtaining higher Tg values for faster deformations and cooler Tg temperatures for slower deformations. Typically, a frequency of 1 Hz is used. This frequency allows for fast data collection at typical ramp rates so that Tg is easily assigned. Consequently, a typical frequency of oscillation of 1Hz is the indicated frequency in standardized tests such as ASTM E1640-13– Standard Test Method for Assignment of the Glass Transition Temperature by Dynamic Mechanical Analysis.

Additionally, it is necessary to select a certain standard for all experiments and methods to obtain Tg values compatible that would allow for the joint use of data of numerous authors to find general composition dependencies. It is well established that Tg values obtained by DMA at a frequency of 1 Hz are compatible with those obtained by Differential Scanning Calorimetry (DSC) for standard cooling/heating rates of 10 K min-1.

  1. DRS measurements were carried out in the temperature range from 20 to 160 degrees. The authors should justify the selected temperature range.

We chose this range to characterize both dipolar and conductive processes, considering the glass transition temperature that DSC evaluated. The lowest temperature is about 50°C below the glass transition temperature evaluated by DSC, and the highest temperature is almost 100°C above the glass transition temperature. Therefore, both dipolar and conductive processes will be visible in our spectrum.

  1. The authors should clarify the adhesion of the coating to the fiber. This has important practical implications. When using coated fiber, some of the coating may come off due to poor adhesion.

In previous studies (refs. [17-19]), we conducted torsion and bending tests, demonstrating that the PEDOT coatings remained stable and adhered to the textile fibres after wear tests. These tests were carried out by monitoring changes in electrical conductivity.

  1. The authors should clarify paragraph lines 147-151. From Table 1 it follows that the decomposition temperature does not decrease for all coatings. A decrease in decomposition temperature leads to a decrease in thermal stability.

We appreciate the suggestion to clarify lines 147-151. The manuscript has been modified accordingly.

"In addition, the decomposition initiation temperature decreases by approximately 100 °C for PEDOT:ClO4, 50 °C for PEDOT:PF6, and 40 °C for PEDOT:BTFMSI compared to the initiation temperature of felt fabric decomposition, as seen in Table 1. When comparing different PEDOT:counterion coatings, it can be observed that PEDOT coatings with ionic liquids as counter ions provide greater thermal stability than the PEDOT:ClO4 coating. Therefore, these fabrics are more stable against higher thermal gradients."

  1. The plasticity of felt fibers does not depend on the ionic liquid on their surface. The authors should clarify lines 188-190

We appreciate the suggestion to clarify lines 188-190. We meant that ionic liquids can act as a kind of 'lubricant' between felt fibres, thus reducing the glass transition temperature.

  1. The statement (lines 220-221) is doubtful. Fiber coating is not taken into account.

In lines 220-221, we are not considering the different coatings on the felt fabric because we are studying the possible microstructure adopted by the felt fibres. The DMA measurement shows two relaxation processes attributed to two glass transition temperatures, which may indicate the presence of two amorphous phases of a different nature, with significantly different chain segment mobility, as per the model in Figure 4 and reference [27]. These relaxation processes are visible in all DMA thermograms, regardless of whether the felt fabrics are coated or not.

  1. Lines 233-235: does this mean the destruction of the MWCNT coating after electrochemical polymerization with various ions.

No, the MWCNT coating remains unaltered after the electrochemical deposition of PEDOT:counterion. Evidence of this is provided by the Raman spectroscopy conducted in previous studies (ref. [19]). The explanation for why the storage modulus decreases after the electrodeposition of PEDOT:counterion is given in lines 237-241.

  1. Graphs 8 and 9 for felt are different. The authors should correct these graphs.

We appreciate the reviewer's comment. The graphics corresponding to figures 8 and 9 (currently 9 and 10) correspond to the same felt sample. The difference between both graphics is that in Figure 10 (previously 9), the axis of electrical conductivity is on a logarithmic scale, while in Figure 9 (previously 8), it is linear. Figure 8, now Figure 9, has been modified by introducing the electrical conductivity axis on a logarithmic scale.

Round 2

Reviewer 2 Report

“Fig. 2” may be "Figure 2" in the revised manuscript.

Author Response

Please see the attached cover letter.

Reviewer 3 Report

The paper can now be recommended for publication

Author Response

Please see the attached cover letter.
